# On the improved stability of the version 7 MIPAS ozone record

Alexandra Laeng[1], Ellen Eckert[1], Thomas von Clarmann[1], Michael Kiefer[1], Daan Hubert[2], Gabriele Stiller[1], Norbert Glatthor[1], Manuel López-Puertas[3], Bernd Funke[3], Udo Grabowski[1], Johannes Plieninger[1], Sylvia Kellmann[1], Andrea Linden[1], Stefan Lossow[1], Arne Babenhauserheide[1], Lucien Froidevaux[4], and Kaley Walker[5]

[1]Institut für Meteorologie und Klimaforschung, Karlsruhe Institute of Technology, Germany
[2]Belgian Institute for Space Aeronomy (BIRA-IASB), Belgium
[3]Instituto de Astrofísica de Andalucía-CSIC, Granada, Spain
[4]Jet Propulsion Laboratory, California Institute of Technology, USA
[5]University of Toronto, Canada

*Correspondence to:* Alexandra Laeng
(alexandra.laeng@kit.edu)

**Abstract.** The Michelson Interferometer for Passive Atmospheric Sounding (MIPAS) was an IR limb emission spectrometer on the Envisat platform. From 2002 to 2012, it performed pole-to-pole measurements during day and night, producing more than 1000 profiles/day. The European Space Agency (ESA) has recently released the new version 7 of Level 1B MIPAS spectra, in which a new set of time-dependent correction coefficients for the non-linearity in the detector response functions was implemented. This change is expected to reduce the long-term drift of the MIPAS Level 2 data. We evaluate the long-term stability of ozone Level 2 data retrieved from MIPAS V7 Level 1B spectra with the IMK/IAA Scientific Level 2 Processor. For this, we compare MIPAS data with ozone measurements from the Microwave Limb Sounder (MLS) instrument on NASA's Aura satellite, ozonesondes and ground-based lidar instruments. The ozonesondes and lidars alone do not allow us to conclude with enough significance that the new version is more stable than the previous one, but a clear improvement in long-term stability is observed in the satellite-data based drift analysis. The results of ozonesondes, lidars and satellite drift analysis are consistent: all indicate that the drifts of the new version are less negative / more positive nearly everywhere above 15 km. The 10-years MIPAS ozone trends calculated from the old and the new data versions are compared. The new trends are closer to old drift-corrected trends than the old uncorrected trends were. From this, we conclude that the non-linearity correction performed on Level 1B data is an improvement. These results indicate that MIPAS data are now even more suited for trend studies, alone or as part of a merged data record.

## 1 Introduction

The Michelson Interferometer for Passive Atmospheric Sounding (MIPAS) was an infra-red (IR) limb emission spectrometer on-board the ENVISAT platform. It performed pole-to-pole measurements during day and night, at altitudes from 6 to 70 km (up to 170 km in special observation modes), providing more than 1000 daily profiles of about 30 species, temperature and cloud composition. In 2002-2004, the instrument operated at full spectral resolution, giving rise to the retrieved ozone product

with a vertical resolution of about 3.5 - 6 km; this period of MIPAS operations is referred to as the full resolution (FR) period. Due to a failure of the instrument's mirror slide in 2004, the operations were suspended during almost a year and were resumed in 2005 with reduced spectral, but improved vertical and horizontal resolution. The corresponding period until the loss of communications with the ENVISAT platform in April 2012, is referred to as the reduced resolution (RR) period of MIPAS operations.

ESA recently released the new version 7 of Level 1B MIPAS spectra. One of two main improvements of this release is that a full instrument misalignment matrix was implemented in this version, which results in better knowledge of tangent altitudes. This change is of minor relevance to the IMK/IAA data product because tangent altitudes are retrieved from the spectra (von Clarmann et al., 2003). Another major improvement is the implementation of a new set of time-dependent correction coefficients for the non-linearity in the detector response functions. In the previous version, the correction coefficients were taken from pre-flight studies and were not time dependent, but the instrument is ageing and the detector response function is changing (Eckert et al., 2014). This improvement of the Level 1B spectra is expected to have a major impact on MIPAS Level 2 data, by reducing the instrument drift. The goal of the present paper is to demonstrate this improvement for the MIPAS IMK/IAA ozone dataset. The MIPAS IMK/IAA dataset versions V7H_O3_40 (2002-2004, for the FR period) and V7R_O3_240 (2005-2012, for the RR period) are part of the new edition of HARMonized dataset of OZone profiles (HARMOZ) database (Sofieva et al., 2013). They are also used in the Long-term Ozone Trends and Uncertainties in the Stratosphere (LOTUS) Stratosphere-troposphere Processes And their Role in Climate (SPARC) Initiative as a parent dataset for two long-term ozone timeseries: the merged SAGE II/MIPAS/OMPS NASA dataset (Laeng et al., 2018) and the merged SAGE II/CCI/OMPS Sask dataset (Sofieva et al., 2017).[1]

## 2  MIPAS retrieval with IMK/IAA research processor

The processing scheme of the MIPAS IMK/IAA research processor, also known as KIT MIPAS Processor, and its adjustment to the RR spectra of MIPAS are described in von Clarmann et al. (2003) and von Clarmann et al. (2009) respectively. In the retrievals, MIPAS Level-1B spectra are inverted to vertical profiles of atmospheric state parameters. After linearization of the radiative transfer problem and after writing the solution in an interative context in order to take nonlinearity into account, the estimation of state parameters is performed iteratively as following:

$$\mathbf{x}_{i+1} = \mathbf{x}_i + \left(\mathbf{K}_i^T \mathbf{S}_y^{-1} \mathbf{K}_i + \mathbf{R} + \lambda \mathbf{I}\right)^{-1} \left(\mathbf{K}_i^T \mathbf{S}_y^{-1} \left[\mathbf{y}_{meas} - \mathbf{y}(\mathbf{x}_i)\right] - \mathbf{R}(\mathbf{x}_i - \mathbf{x}_a)\right).$$

Here:

- $\mathbf{x}_i$ is the $\mathrm{n}_{max}$-dimensional vector of unknown parameters estimated on $i$-th iteration

- $\mathbf{K}_i$ is the $\mathrm{m}_{max} \times \mathrm{n}_{max}$ Jacobian, containing the partial derivatives of all $\mathrm{m}_{max}$ simulated measurements $\mathbf{y}$ under consideration with respect to all unknown parameters $\mathbf{x}$.

---

[1]SAGE II: Stratospheric Aerosol and Gas Experiment, CCI: ESA's Climate Change Initiative, OMPS: Ozone Mapping and Profiler Suite, NASA: National Aeronautics and Space Administration

- $\mathbf{K}_i^T$ denotes the transposed matrix $\mathbf{K}_i$,

- $\mathbf{S_y}$ is $\mathrm{m}_{max} \times \mathrm{m}_{max}$ covariance matrix of measurement noise,

- $\mathbf{R}$ is $\mathrm{n}_{max} \times \mathrm{n}_{max}$ regularization matrix

- $\lambda\mathbf{I}$ is $\mathrm{n}_{max} \times \mathrm{n}_{max}$ Levenberg-Marquardt term (Levenberg, 1944; Marquardt, 1963)

- $\mathbf{y}_{meas}$ is the $\mathrm{m}_{max}$-dimensional vector of measurements under consideration,

- $\mathbf{y}(\mathbf{x}_i)$ is the forward modeled spectrum using parameters $\mathbf{x}_i$ from the $i$-th iteration step.

- $\mathbf{x}_a$ is the related a priori information

In stratospheric/tropospheric retrievals, local thermodynamic equilibrium (LTE) is assumed (the processor is designed such that the major contributors to the infrared spectrum are the first to be retrieved, before the gases with weak spectral features).
First, the spectral shift of the spectra is determined. Then, temperatures and altitude pointing information (i.e. the elevation angle of the line of sight of the instrument) are jointly retrieved; the retrieved quantity is referred to as TLOS. Then, traces gas abundances are retrieved. The sequence of these operations in V7 retrievals is: $O_3$, $H_2O$, and then the other trace gases. As a general rule, results of preceding steps are used as input for the subsequent retrieval steps, i.e. the $O_3$ retrieval uses retrieved temperatures and pointing information, and the subsequent trace gas retrievals will use retrieved $O_3$ abundances. Beside each
target species, microwindow-dependent continuum radiation profiles and microwindow-dependent, but height-independent, zero level calibration corrections are jointly fitted.

## 2.1 New V7-dedicated ozone retrieval setup

A new dedicated ozone retrieval setup was recently developed for the IMK/IAA Level 2 processor. The MIPAS ozone profiles produced with this retrieval setup are referred to as data versions V7H_O3_40 and V7R_O3_240. All improvements reported
below refer both to the FR and the RR data.

One of the most relevant changes in the improved retrieval setup is not related to the ozone retrieval itself but to the preceding temperature retrieval. The radiative transfer forward calculation now uses information on the horizontal two-dimensional temperature variation along-line-of-sight from the European Centre for Medium-Range Weather Forecasts (ECMWF) re-analysis. While the temperature at the measurement geolocation is still retrieved from the MIPAS spectra, the horizontal temperature
variation is prescribed by the analysis data. This allows a more accurate retrieval in situations where neither the assumption of local horizontal uniformity nor the approximation of horizontal temperature variation by linear gradients, as described by von Clarmann et al. (2009), is adequate. Typically the old approaches lead to problems when the line of sight of a measurement crosses the edge of the polar vortex. We refer to this change as to 2D T-field. Above approximately 60 km there is no ECMWF data available to build the 2D-information. There, a retrieval of a linear horizontal gradient, in addition to the temper-
ature retrieval, is performed. However the retrieved gradient profile is strongly regularized towards zero below the stratopause.

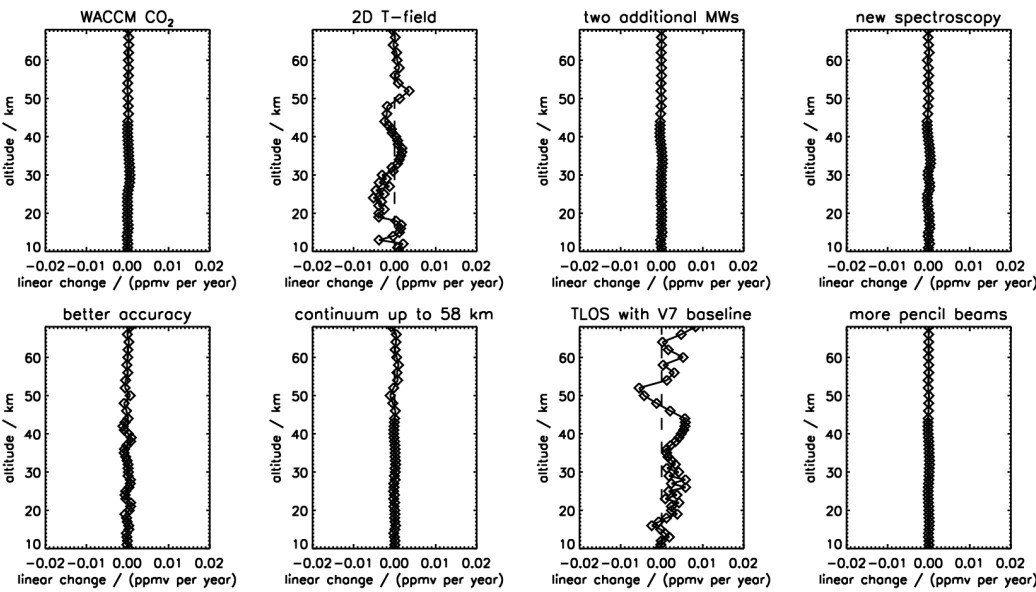

**Figure 1.** Sensitivity study : how different changes in retrieval setup affect the ozone retrieval. For each change, we show the linear term of differences of ozone retrieved from V7 spectra, using the new retrieval setup and the previous retrieval setup. Each panel correspond to a retrieval setting where only one parameter (which is stated on the title of the panel) is changed, and all other parameters remain unchanged.

Only above it serves to include gradient information at altitudes where these are not given externally. All these changes in the temperature retrieval are referred to as TLOS V7 retrieval.

Above the uppermost tangent altitude, MIPAS cannot measure altitude-resolved temperature profiles, because at most one degree of freedom is achievable there. Thus, the choice of adequate temperature a priori information is particularly important

at these altitudes. In previous data versions, MSIS (Hedin, 1991) temperatures served as a priori at these altitudes. Particularly in situations of elevated stratopause events, this climatological model did not represent the actual upper stratospheric/lower mesospheric conditions very well, and the related temperature retrieval error was found to propagate noticeably onto the retrieved ozone. Validation results of the previous version were found to be particularly biased during such elevated stratopause events. Thus, in this new version, a priori information on temperature at altitudes above 60 km is taken from a dedicated

temperature climatology based on simulations of the Whole Atmosphere Community Climate Model (WACCM) and MIPAS middle and upper atmospheric modes temperatures from the V5 version (García-Comas et al., 2014). The WACCM temperature fields were taken from CCMI-REFC1SD simulations, in specified dynamics mode, sampled at MIPAS locations and times. The WACCM model is described in detail by (Marsh, 2011; Marsh et al., 2013), while the implementation of the specified dynamics mode and the most recent changes in the model, including those of the gravity wave parameterization and the Prandtl number,

are described in (García et al., 2016) and López-Puertas et al. (2017). In the a priori temperature climatology, the WACCM

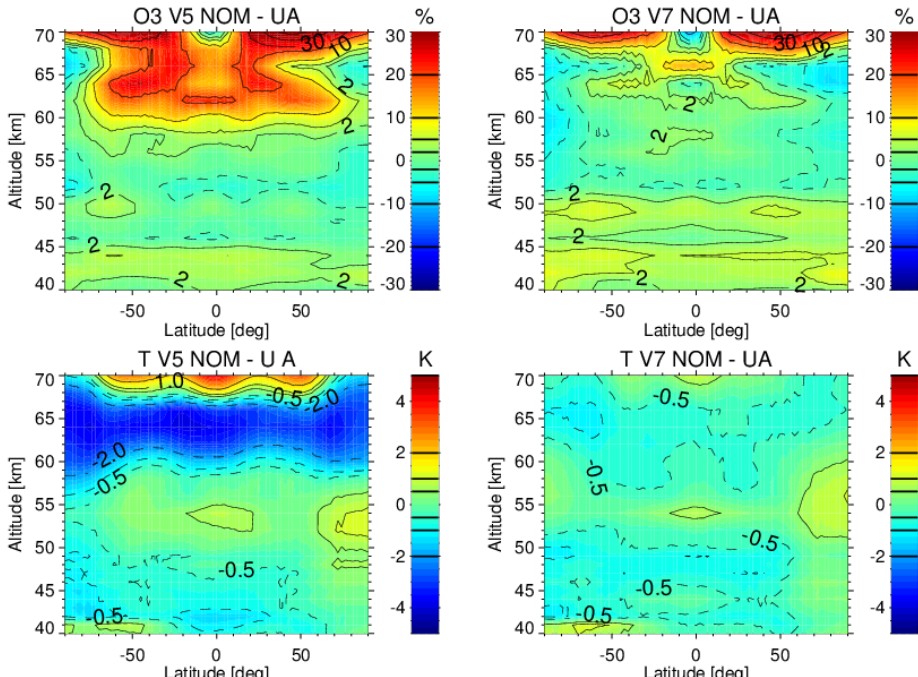

**Figure 2.** Climatological mean differences of ozone (top row) and temperature (bottom row) retrievals from the old (left column) and new (rigt column) version of data to retrievals from MIPAS middle and upper stratopheric modes.

temperature fields have been corrected by using the MIPAS middle and upper atmosphere temperatures (García-Comas et al., 2014). For this, an altitude and latitude dependent seasonal correction function has been derived from multi-annual averages of MIPAS-collocated WACCM differences. Also the $CO_2$ mixing ratios used for the temperature retrieval are now based on WACCM simulations.

To evaluate the effect of the new (V7) retrieved temperature on the V7 NOM retrieved $O_3$, we compared i climatological means of temperature and ozone from the old and new MIPAS data versions with MIPAS retrievals from middle and upper atmosphere modes. The latter data were already validated for both temperature (García-Comas et al., 2014) and for $O_3$ (López-Puertas et al., 2018). Figure 2 shows climatological mean differences in ozone (top row) and temperature (bottom row) for the version 5 (old) (left column) and version 7 (new) (right column). A close look at the 55-67 km altitude region in Figure 2
reveals that the use of the new V7 temperatures improves substantially the agreement in low mesosphere of both temperature and ozone retrieved from MIPAS nominal mode with respect to MIPAS middle and upper atmosphere modes.

Further changes in the $O_3$ retrieval setup refer to the background continuum radiation. Initially, as described in von Clarmann et al. (2003), in retrievals of temperature, ozone, and of most of subsequent species, the continuum absorption coefficient was fitted up to 32 km altitude only. Fitting background continuum radiation was deemed not necessary at upper altitudes, because
no sizable aerosol contributions were expected above the stratospheric aerosol layer, which extends from the tropopause up

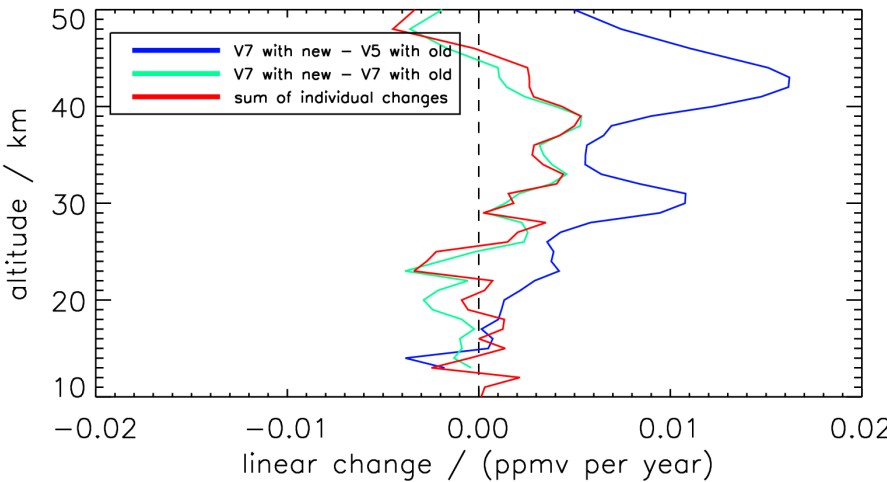

**Figure 3.** Sensitivity of the retrieved ozone trend to the changes in retrieval setup and Level 1B spectra. V5 and V7 refer to the versions of Level 1B MIPAS spectra on which the retrieval is run. "Old" and "new" refers to the old and new retrieval setups. The blue line shows the linear change in ozone in case when both the retrieal setup and the version of the Level 1B spectra are changed. The green line illustrates the linear change in ozone in case when only the retrieval setup is changed, and applied in both cases on the same Version 7 Level 1B spectra. The red line shows the linear change in ozone originating from the sum of individual changes in the retrieval, shown in Figure 1, applied to the same version of Level 1B spectra.

to 30 km altitude. However, Neely III et al. (2011) provided some evidence that, due to meteoric dust, there are relevant aerosol particles abundances above the stratospheric aerosol layer. Background continuum radiation was then fitted up to 48 km altitudes for the previous versions of ozone retrievals (V5R_O3_224/225) and for some subsequent species. For $SF_6$ (continuum fitted up to 49 km) and $CH_4$ and $N_2O$ (continuum fitted up to 58 km) retrievals, this approach removed some
retrieval artefacts (Haenel et al., 2015; Plieninger et al., 2015). In order to homogenize the retrievals of different species, in the present version, both temperature and ozone retrievals contains background radiation fitted up to 58 km altitude. In the new retrieval settings, $H_2O$ mixing ratios are treated as a joint fit variable in the ozone retrieval. For this purpose two additional $H_2O$ microwindows have been included in the ozone retrieval and $H_2O$ is jointly fitted with ozone. This provides a better constraint of $H_2O$ in the upper troposphere and lower stratosphere (UTLS).
Beyond this, the forward model KOPRA (Stiller, 2000) is now run with higher numerical accuracy. The change consists in a somewhat smaller wavenumber interval for calculation of the absorption coefficients, (from $0.00125$ cm$^{-1}$ to $0.009765625$ cm$^{-1}$), and in an extended width (up to a decrease to 0.1% of center value) of the applied apodisation function. In the retrieval of TLOS, preceding the retrieval of ozone, now 5 pencil beams are used for the numerical integration of radiances over the instrument fields of view for all tangent altitudes. While the same $O_3$ spectroscopic data were used as with preceeding data
versions, spectroscopic data of some interfering species were updated, e.g.: $CO_2$ spectroscopy was updated from MIPAS pf3.0 (Flaud et al., 2003) to HITRAN-2012; $N_2O$ spectroscopy was updated from HITRAN-2000 to HITRAN-2008; $CH_4$

spectroscopy was updated from 2002-update of HITRAN-2000 to HITRAN-2008; OCS spectroscopy was updated from 2006-update of HITRAN-2004 to 2009-update of HITRAN-2008.

The effect of changes in the retrieval setup, summarized above, on the estimated trends were analysed in the following way. The retrieval was applied to the V7 MIPAS spectra first with old setup. Changes described above were then applied one by one, while all other changes were desactivated. For each change, differences in ozone retrieved with the old setup and with the setup modified by one parameter only, were calculated for a set of orbits over MIPAS Reduced Resolution period, then the linear term of differences was extracted, in ppmv per year. Results of this sensitivity study are presented in Figure 1. As expected, two major impacts on ozone come from the way the temperature is retrieved in a preceding step. The sum of all individual changes is plotted as red curve in Figure 3. The difference between the estimated ozone trends with a retrieval setup in which all individual changes are applied simultaneously, and the trends estimated with old setup, both applied to the same new MIPAS version 7 spectra, is plotted as green curve in Figure 3. As expected, it is very close to the sum of inidividual changes in the retrieval setup (red curve). The total change in ozone retrieval (new spectra and new retrieval setup) is plotted as blue curve in Figure 3. As expected, the major cause of the total change of retrieved ozone is the use of the new version of MIPAS spectra; it can be visualized as the difference beween the blue and the green curves in Figure 3.

## 2.2 Diagnostics

IMK-IAA MIPAS results are characterized by error estimates, as well as vertical and horizontal averaging kernels. The latter two are used to estimate the spatial resolution of the retrievals. The gain function is calculated as follows:

$$\mathbf{G} = (\mathbf{K}^T \mathbf{S}_y^{-1} \mathbf{K} + \mathbf{R})^{-1} \mathbf{K}^T \mathbf{S}_y^{-1}$$

The covariance matrices of the state vector and of the measurement are linked by $\mathbf{S}_x = \mathbf{G} \mathbf{S}_y \mathbf{G}^T$. The averaging kernel matrix, reflecting the sensitivity of the retrieved profile to the change of state parameters, is $\mathbf{A} = \mathbf{G}\mathbf{K}$. The random error covariance matrix $\mathbf{S}_{\mathrm{random}}$ of the retrieved quantity x is calculated as

$$\mathbf{S}_{\mathrm{random}} = \left(\mathbf{K}^T \mathbf{S}_{\mathbf{y}}^{-1} \mathbf{K} + \mathbf{R}\right)^{-1} \mathbf{K}^T \mathbf{S}_{\mathbf{y}}^{-1} \mathbf{K} \left(\mathbf{K}^T \mathbf{S}_{\mathbf{y}}^{-1} \mathbf{K} + \mathbf{R}\right)^{-1}$$

and the linear mapping $\Delta \mathbf{x}_j$ of the uncertainty $\Delta b_j$ of parameter $b_j$ is

$$\Delta \mathbf{x}_j = (\mathbf{K}^T \mathbf{S}_y^{-1} \mathbf{K} + \mathbf{R})^{-1} \mathbf{K}^T \mathbf{S}_y^{-1} \times [\mathbf{y}(\mathbf{x}, b_j + \Delta b_j) - \mathbf{y}(\mathbf{x}, b_j)].$$

The covariance matrices are stored for all retrieved profiles and are available on demand. Two additional criteria are usually applied to the retrieved data in order to evaluate the quality of the profile: 1) results where the diagonal value of averaging kernel is less (in absolute value) than 0.03 are considered non-trustful; 2) results related to parts of the atmosphere not sensed by MIPAS (i.e. below the lowermost used tangent altitude) are considered non-trustful. The is no substantial change in averaging kernels of retrieved ozone profiles between version 5 and version 7. The vertical resolution is determined by the vertical sampling of the instrument, which in case of MIPAS RR measurements varied from 1.5 to 4 km, and regularization. The vertical resolution varies between 2.5 - 4 km at 10 - 40 km altitude, and 4 - 6 km at 40 km and higher. The vertical resolution

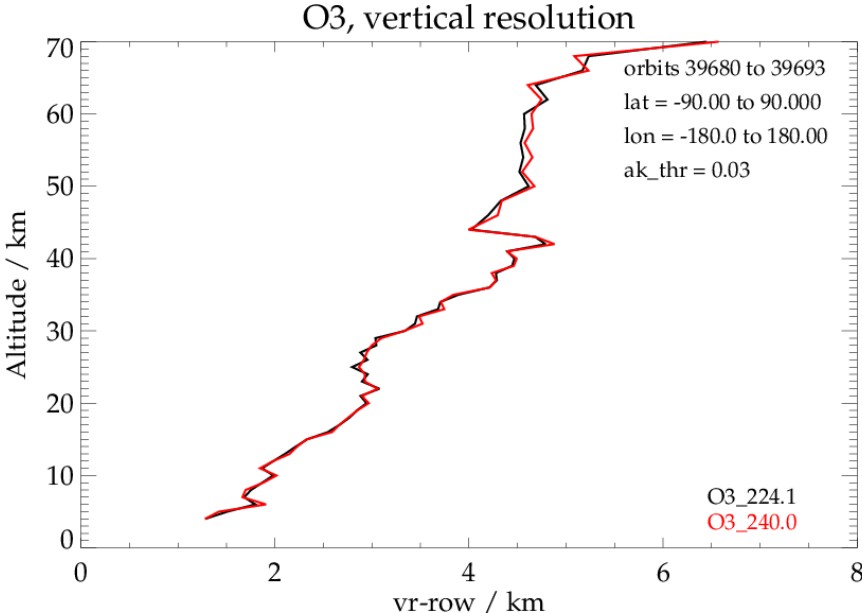

**Figure 4.** Vertical resolution of ozone profiles from versions V5 and V7, derived from the full-width at half maximum (FWHM) of rows of averaging kernel matrix.

derived from the full-width at half maximum (FWHM) of rows of averaging kernel matrix for the old and the new version is shown in Figure 4: it varies between 2 and 3 km in the troposphere, 3-5 km in the stratosphere, and goes up to 5 km at 50-65 km altitudes. The vertical resolution did not changed substantially between the two versions.

### 2.3 Sanity check of the output of the new processing scheme

In this section we check whether the new processing scheme and new MIPAS Level 1B spectra introduced artificial biases in the resulting ozone profiles. For this, we compare the MIPAS ozone profiles from the new and the old versions with ozone profiles measured by ozonesondes launched from two ground stations, and ozone profiles measured by two satellite instruments. The two MIPAS datasets from the FR and RR periods must be treated as independent datasets, because of differences in the processing set-ups and different vertical resolutions coming from different tangent altitude patterns. We work with two versions

of the MIPAS RR (2005–2012) ozone data measured in the instrument nominal mode: the version V5R_O3_224/225, which we refer to as "the old version", and the version V7R_O3_240, which we refer to as "the new version". The old version was validated in Laeng et al. (2014). This section does not aim to be a full validation of the new version; the latter was performed in the frame of ESA Ozone_cci project and will be presented in the upcoming paper (Hubert et al., 2018).

The ozone profiles from the two versions were compared to ozonesonde profiles launched from two stations: mid-latitudinal

Boulder at 39.99°N and sub-tropical Hilo at 19.72°N. A more extensive ground-based comparison of V7R_O3_240 will be

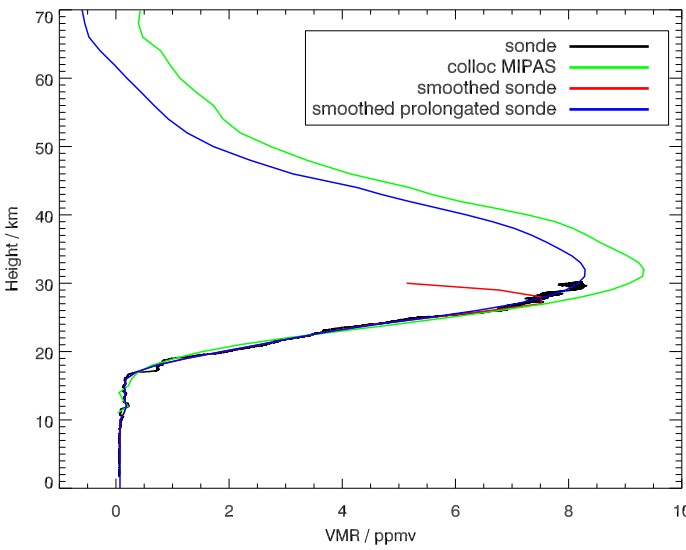

**Figure 5.** Application of MIPAS AK to the ozonesonde profiles: original ozonesonde profile, collocated MIPAS profile at geolocation 20100630T164035Z, and interpolated, smoothed and prolongated ozonesonde profiles .

presented in the upcoming paper of Hubert et al. (2018). We have chosen the collocation criteria of 1000 km and 24 hours, which gave rise to 1184 collocated profiles for Boulder station and 1188 collocated profiles for Hilo station. In order to take into account the differences in vertical resolutions between ozonesondes and MIPAS profiles, the ozonesonde profiles were smoothed with the MIPAS averaging kernels. When applying the averaging kernels to ozonesonde profiles, the boundary

effects are important. To reduce this, each ozonesonde profile was extended at heights over 30 km by a shifted collocated profile from MIPAS, see Figure 5.

The results of this intercomparison are shown in Figure 6. The percentage relative bias with respect to the reference instrument REF, $100\times$(MIPAS-REF)/REF of the old MIPAS version is shown in pink, and the new version in green. The results from both stations look fairly consistent, indicating an agreement within $\pm$ 5% at 20-30 km height, with the new version

having a slightly larger (up to 1%) bias with respect to ozonesonde profiles compared to the old version in this height range. A clear improvement can be observed at 12-14 km, with the bias reduced by 5% for Boulder and by up to 10% reduction at Hilo. Moreover, for the Hilo station, a bias reduction from 15% to 5% from 12 to 20 km can be observed. All these biases are significant at the $2\sigma$ Level.

To assess the bias of the new MIPAS dataset with respect to satellite reference retrievals, the previous and the new version of

MIPAS are compared to the ACE-FTS and MLS datasets. For consistency, both the old and the new versions were compared to the same version of reference instruments, the version 3.5/3.6 of ACE-FTS and the version 3.3 of MLS. The version 3.3 of MLS data was used instead of the most recent version 4.2 in order to keep consistency with the analysis of Eckert et al.

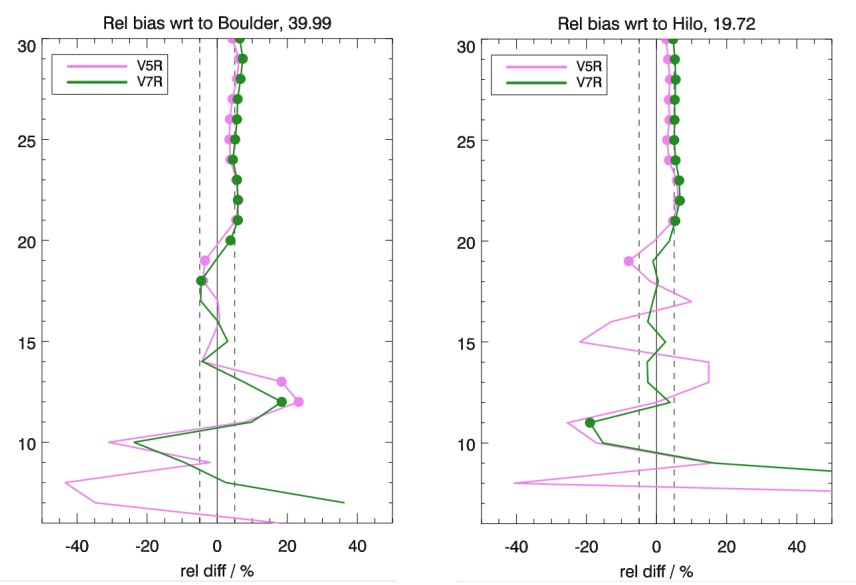

**Figure 6.** Biases with respect to Boulder (39.99N) and Hilo (19.72N) ozonesondes for the old (V5R) and new (V7R) MIPAS data versions. The altitudes where the bias is significant at $2\sigma$ level are highlighted by circles.

(2014). To avoid any sampling effects, comparisons used exactly the same coincidences between the reference data set and the old and new MIPAS data. The coincidence criteria were chosen to be 500 km and 5 hours for ACE-FTS, and 250 km and 4 hours for MLS. This gave rise to ~345000 matched pairs for each version of MIPAS with MLS, and ~8300 matched pairs for ACE-FTS. Also here, we applied the averaging kernels of MIPAS to both satellite reference datasets; the effect of this operation

is, however, negligible, due to the small contrast in vertical resolutions of the three datasets. Figure 7 shows the results of these comparisons. The error bars show the standard deviation of correspoding differences. We do not show the standard errors of the means because, due to large sample sizes, the values of standar errors of the means are too small in this case and are not be visible in Figure. We choose to present the global average; zonal averages were compared as well (not shown here), no latitudinal biases were cancelled out when going to the gloabl average. With respect to ACE-FTS, the new version of MIPAS

ozone dataset results in a reduction of the bias by about 2% at 10-20 km, 1 to 4% at 30-52 km, and by about 6-8% at 60 km and higher. With respect to MLS, a reduction of the bias by 1 to 12 % can be observed in the lower mesosphere (52 km and higher), while at other heights the bias is slightly increased by 1 to 4%. Historically, around the ozone vmr peak, MIPAS ozone measurements tended to have a high bias (Laeng et al., 2015), which is attributed to the use of the microwindows in the MIPAS AB spectral band (1020 - 1170 cm$^{-1}$) at these heights (Glatthor et al., 2018). In this height region, around 35 km, the

new version demonstrates an improvement with respect to ACE-FTS where the bias is reduced from 2% to almost 0%. With respect to MLS, the bias has increased slightly from 3% for the previous version to 5% for the new version. In the UTLS, the comparison with ACE-FTS still demonstrates a clear improvement, and MLS compares better with the new version at 10-12 km and 14-16 km.

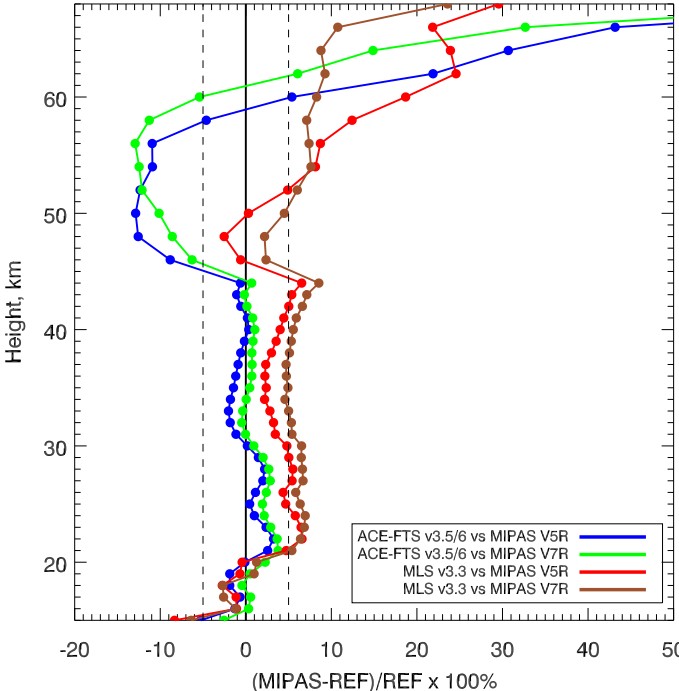

**Figure 7.** Bias between old and new verions of MIPAS ozone data and the ACE-FTS and MLS profiles. The altitudes were the bias is significant at $2\sigma$-level are highlighted by circles. The averages over the whole globe are presented.

From ozonesonde and satellite comparisons described above, we conclude that no artificial biases were instroduced neither by the new MIK/IAA MIPAS processing scheme nor by the new version of MIPAS Level 1B spectra. This conlcusion is consistent with findings of the validation activities within ESA Ozone_cci project (Hubert et al., 2018).

## 3 Drift estimation

### 3.1 Level 1B analysis: improvement in the detector non-linearity characterisation

The MIPAS instrument recorded interferograms in the five spectral bands, A: 685 - 970 $cm^{-1}$, AB: 1020 - 1170 $cm^{-1}$, B: 1215 - 1500 $cm^{-1}$, C: 1570 - 1750 $cm^{-1}$, and D: 1820 - 2410 $cm^{-1}$, using 8 infra-red detectors called A1/A2, B1/B2, C1/C2, and D1/D2. The spectral signal in each spectral band is composed of the combined information from the different detectors as follows: the A1/A2 detectors contribute to the MIPAS spectral band A, B1 to AB, B2 to B, C1/C2 to C, and D1/D2 to D. Four detectors, namely A1/A2 and B1/B2, show a non-linear response to photon flux which has to be corrected to get the appropriate interferogram. The detectors exhibited ageing; essentially the sensitivity degraded slowly over time and the response became more linear. Hence, for the spectral bands A, AB, and B, precisely those relevant for the temperature and ozone retrieval, an impact of the detector ageing on the measurements has to be expected. There was just one pre-flight characterization

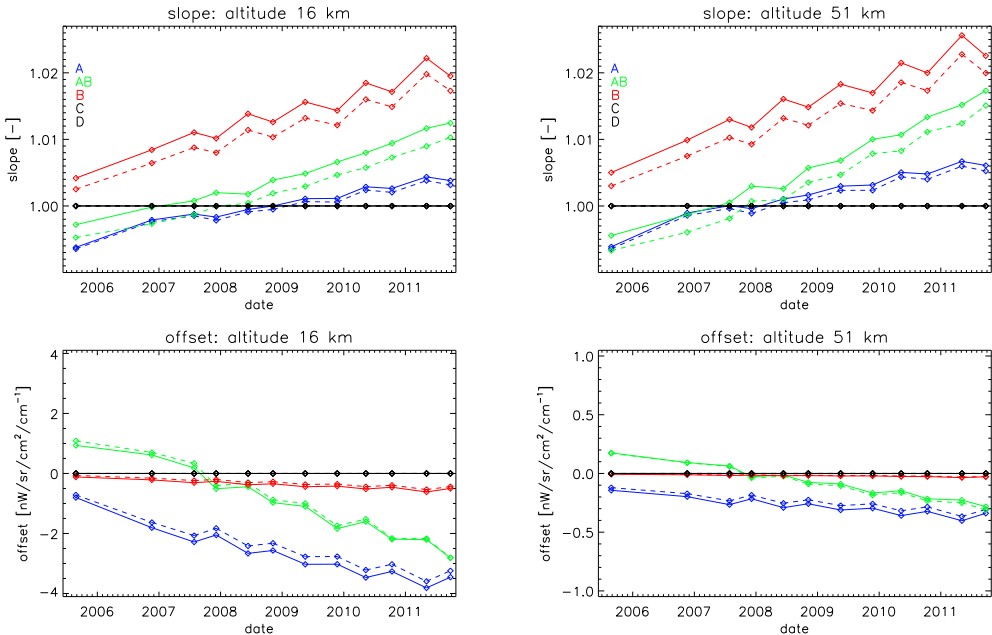

**Figure 8.** Slope and offset of the linear fit of MIPAS RR Level 1B Spectra with the old set of NL coefficient versus spectra with new set of NL coefficients at 16 km (left) and 51 km (right). Solid lines correspond to forward mirror movement, and dashed lines correspond to reverse mirror movement.

measurement for the detector non-linearity (NL). This characterization was used for inflight gain calibration throughout the mission (2002-2012) to correct for non-linearity in detector responses up to version 5 of Level 1B MIPAS data. This deficient radiometric calibration caused a drift of the instrument (Kleinert et al., 2007; Kiefer et al., 2013; Eckert et al., 2014), making MIPAS data not suitable for trend studies without preliminary drift removal. Wagner and Birk (2005) proposed a new method

5    to characterize the detector non-linearity from in-flight measurements in raw data mode. This correction was incorporated in version 7 of MIPAS Level 1B Spectra. Plotting MIPAS Level 1B spectra with new non-linearity coefficients (NLC) against spectra with old NLC allows to fit two regression parameters: slope and offset, with ideal values to be close, respectively, to 1 and 0. These slopes and offsets, for both directions of the interferometer mirror movement, forward and backward, are shown at the Figure 8 for altitudes of 16 km (left panel) and 51 km (right panel).

10    Figure 8 demonstrates that for Level 1b MIPAS spectra in bands A, AB, and B, both slope and offset change over time, i.e. the spectra with the old NLC and the new NLC slowly drift apart (up to 2%). This is the reason why an improvement in long-term stability of MIPAS Level 2 data is expected.

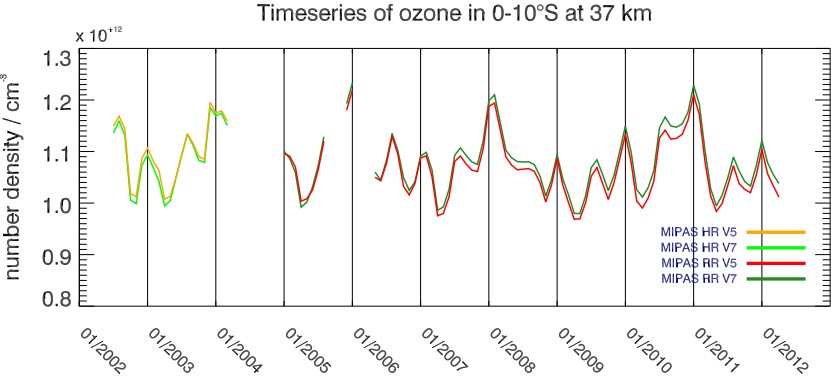

**Figure 9.** MIPAS timeseries in 0-10°S latitude band at 37 km. Drift has clearly linear structure, which is an indirecte evidence that the methods of estimating MIPAS drift, choosen in Eckert et al. (2014) are appropriate.

## 3.2 Level 2 analysis

In our definition, drifts are long-term variations of the bias between two instruments. The drifts appear as artificial trends of a signal due to imperfect instrument stability. To assess the long-term stability of the MIPAS ozone dataset, we compared it to a network of data including ozonesondes and ground-based lidars using the method from (Hubert et al., 2016), and with Aura
MLS (Froidevaux et al., 2008), using the method described by Eckert et al. (2014). As a first rough estimation, we examine the timeseries of old and new versions of MIPAS ozone data at 37 km altitude in 0-10°S latitude band, see the Figure 9. The choice of the latitude band and altitude for such an illustration are driven by the drift estimation from Eckert et al. (2014), see also the left panel of the Figure 11: it corresponds to a pixel with strong significant negative drift. The red and the dark green curves in Figure 9 show the timeseries of the old and the new MIPAS RR data respectively. A negative drift of the old data
with respect to the new data is clearly seen and is clearly linear. This provides indirect evidence that the methods of estimating the drift of MIPAS, chosen in (Hubert et al., 2016) and (Eckert et al., 2014), are appropriate.

### 3.2.1 Drift with respect to ground-based instruments

We calculated the network-averaged drift of the old and new MIPAS data versions versus co-located lidar and ozonesonde ozone profiles using the regression and averaging procedure described in Hubert et al. (2016). The ground-based ozonesonde
and lidar networks provide, at numerous sites around the world, vertical ozone profiles that –together– cover regions of the troposphere and stratosphere. The sonde data were taken from the archives of the World Ozone and UV radiation Data Center (WOUDC, http://www.woudc.org), the Network for the Detection of Atmospheric Composition Change (NDACC, http://www.ndacc.org, Kurylo & Zander (2001); Lambert et al. (1999)) and the Southern Hemisphere Additional Ozonesondes (SHADOZ, http://croc.gsfc.nasa.gov/shadoz, Thompson et al. (2012)). The lidar data were taken from the NDACC archive
(http://www.ndacc.org). The MIPAS profiles were compared to any ground-based measurements that co-locate within 6 hours and 500 km. The latter were smoothed to the vertical resolution of MIPAS and regridded to its native vertical grid. The bias of

MIPAS (in %) is estimated as the median value of the relative difference time series $\Delta x_j(t_i,l) = 100 \times \frac{x_{j,sat}(t_i,l) - x_{j,gnd}(t_i,l)}{x_{j,gnd}(t_i,l)}$, where $x_{j,sat}(i,l)$ and $x_{j,gnd}(i,l)$ represent, respectively, satellite and (smoothed and regridded) ground-based ozone at grid Level $l$ of co-located profile pair $i$ at time $t_i$ and station $j$. The choice by Hubert et al. (2016) to work in relative units was mainly made to divide out any multiplicative time dependence (e.g., seasonal cycle). This makes the difference values sensitive to low

ozone concentrations (e.g., in the UTLS). The analysis of the long-term stability of MIPAS data is performed in two steps. First, the linear drift is estimated at each ground station and vertical Level, using an iterative Tukey-bisquare reweighted least-squares procedure to fit the relative difference time series to a linear regression model $\Delta x_j(t,l) = \alpha_j(l)(t - t_0) + \beta_j(l) + e_j(t,l)$. In a second step, these drift estimates are averaged over a number of ground stations to obtain the network-averaged drift of MIPAS (in % per decade, see Hubert et al. (2016) for more details). The averaging reduces the uncertainty from spatial and temporal

inhomogeneities present in the ground-based data sets; more details can be found in Hubert et al. (2017).

     The resulting network-averaged drift calculated for the old and new MIPAS data versions versus lidar and ozonesondes data are presented in Figure 10. Ozonesonde and lidar results are fairly consistent. The MIPAS drift relative to ground-based instruments is not significant for either version. Over the altitude range of 15-37 km, the drift of the old version is 1% per decade ( 2% at the most) more negative / less positive (depending on altitude) than the drift of the new version. The difference

in the drifts between the old and the new version is not significant because the estimated $2\sigma$ uncertainty in the drift is at best 2%/decade between 20-25 km, and larger elsewhere. Hence, the ozonesonde and lidar analyses alone do not allow us to conclude with certainty which data version is more stable. But still this analysis indicates that it is most likely that the new MIPAS data exhibit a smaller negative drift.

### 3.2.2   Drift with respect to Aura MLS

For calculating drifts with respect to satellite instruments, we have chosen the Aura MLS data as reference dataset. The reason is that MLS is a dense sampler, in a relative sense, at some heights, MIPAS time series do track the MLS time series more closely, and it is known to be stable (Hubert et al., 2016). The MIPAS drift estimation with respect to MLS uses profiles matched to 250 km and 6 hours, in the period 2005–2012. The differences between the MIPAS and MLS measurements were taken at every valid altitude grid point of each profile pair; then monthly zonal means of these differences were calculated in

10-degrees latitude bins. The multi-linear regression model, described in Eckert et al. (2014) and von Clarmann et al. (2010), was then applied to the timeseries. The resulting drifts are shown in the right panel of Figure 11 as a function of altitude and latitude band. The left panel of Figure 11 shows the same drift estimates for the previous version of MIPAS data. Hatching indicates domains with less than $2\sigma$ significance.

     The comparison of the old (left) and the new (right) version in Figure 11 reveals the following features. In the new version,

there are a lot fewer areas with significant drift. Combined with the fact that drift uncertainties are of similar size for the old and new version, we interpret this reduction of significant areas as an improvement: the drifts became smaller in absolute values. As pointed out by Eckert et al. (2014), the old version exhibited mostly negative drifts, going down to -0.33 ppmv/decade and becoming more negative with altitudes up to $\sim$ 40 km. This clear pattern of significant negative drifts was extending over all latitudes at altitudes over 30 km, and going down to 20 km at mid-latitudes. In the new version, this pattern has disappeared

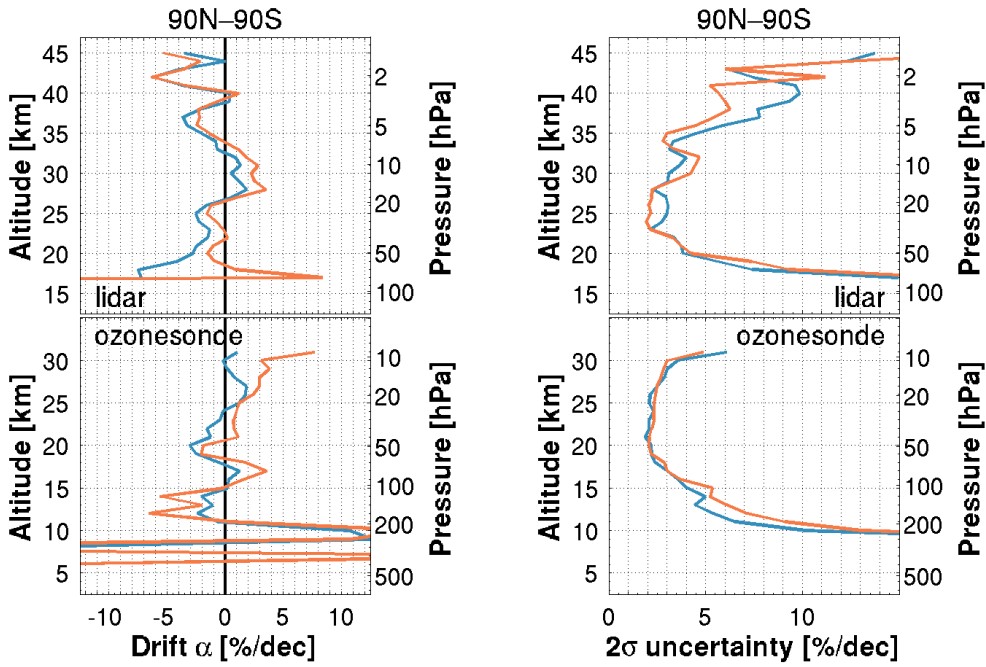

**Figure 10.** Left panel: network-averaged drift of the old (blue) and the new (orange) MIPAS ozone versus lidar (top) and ozonesonde (bottom) measurements for 2005-2012 period profiles. Right panel : 2$\sigma$ uncertainty of estimated drifts.

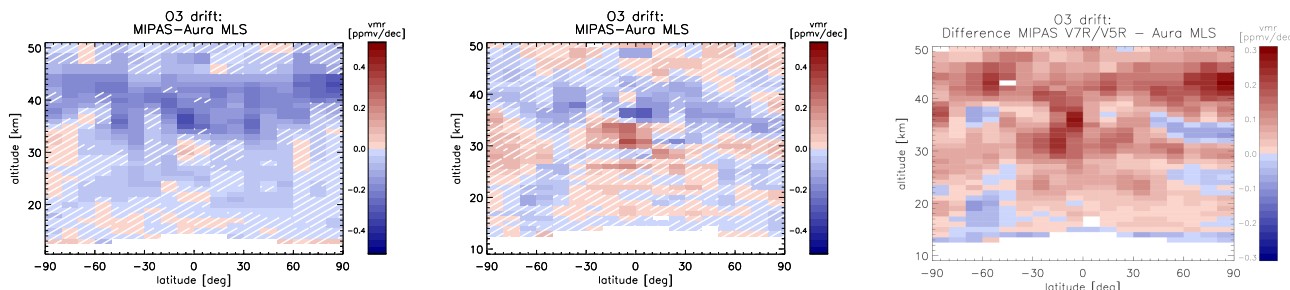

**Figure 11.** Altitude-latitude cross-section of absolute drifts of MIPAS V5R (left) and V7R (middle) vs. Aura MLS ozone measurements as well as their difference (right). Hatched areas mean that the significance is less than 2 sigma.

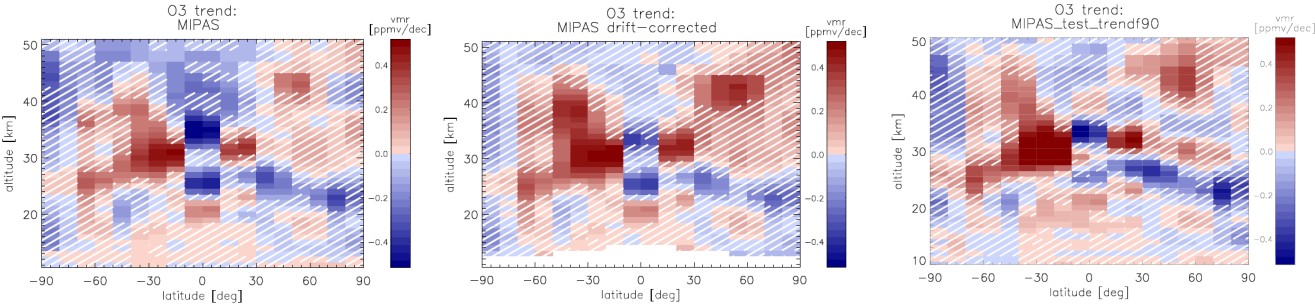

**Figure 12.** MIPAS 10-years trends calculated based on V5R data without drift correction (left), drift-corrected V5R data (middle), and V7R data without drift correction (right). Hatched areas mean that the significance is less than 2 sigma.

and there is no clear systematic drift pattern; some areas with small positive drifts are neighboring areas with small negative drifts, with the majority of both areas being non-significant. This can be attributed to the small absolute values of the drifts. The appearance of some pixels with significant positive drifts is tentatively attributed to an over-correction in non-linearity coefficients in Level 1B spectra. The version 8 of Level 1B MIPAS spectra, which at the time of this writing is in a testing
phase, is expected to correct for this undesirable effect.

In summary, out that the ground-based and MLS-based analysis provide consistent results. Both indicate that the drifts of the new version are less negative / more positive nearly everywhere above 15 km.

## 4 Update of 10-year trends from MIPAS

Although 10 years of data provides a timeseries which is too short to sample the solar cycle, we calculated the newly estimated
trends on it, in order to compare with similar trends calculated in Eckert et al. (2014). The comparison of MIPAS 10-years calculated on old and new datasets are shown in the Figure 12. For convenience of the reader, we reproduce two relevant figures from Eckert et al. (2014) as the left and middle panels of the Figure 12.

One should be aware that trends in Eckert et al. (2014) are already drift-corrected, so we do not expect an improvement of final drift-corrected trend estimates from Eckert et al. (2014). Instead, we expect that the new trends (right panel of the
Figure 12) are closer to drift corrected trends from Eckert's paper (middle panel in the Figure below, this is Fig. 15 from Eckert's paper) than uncorrected original trends were (left panel in the Figure 12, this is Fig. 13 in Eckert et al. (2014)). As this expectation is confirmed, we learn that the non-linearity correction is successful and probably also applicable to other species where no easy drift-correction via comparison to external instruments is possible.

A time series of 10 years is arguably too short to estimate a climatologically relevant trend, a longer timeseries can be
constructed by merging MIPAS ozone record with SAGE II and OMPS ozone records (Laeng et al., 2018).

## 5  Conclusions

We have presented a new retrieval setup for MIPAS $O_3$ retrieval and assessed the long-term stability of the new MIPAS ozone dataset retrieved with KIT IMK/IAA Scientific Level 2 MIPAS Processor from the new version 7 Level 1B MIPAS spectra. As a sanity check of the new retrieval scheme, we assessed the bias of the old and of the new datasets with respect to two ozonesonde stations, Boulder and Hilo, and two satellite instruments, ACE-FTS and MLS. The biases with respect to the ozonesondes from Hilo and Boulder stations are consistent, with an agreement within $\pm 5\%$ at 20-30 km. At 12-14 km, the bias against Boulder is reduced by 5%. For the Hilo station, the bias is reduced from 15% to 5% from 12 to 20 km. The old and new MIPAS datasets were compared with ACE-FTS and MLS satellite instruments. With respect to ACE-FTS, the new version of the MIPAS ozone dataset shows a bias reduction by about 2% at 10-20 km, a bias reduction by about 1 to 4% at 30-52 km, and a bias reduction by about 6-8% at 60 km and higher. With respect to MLS, a bias reduction by 1 to 12 % is observed in the lower mesosphere (52 km and higher), while at other heights the bias is slightly increased by 1 to 4%. Around 35 km, the new version demonstrates an improvement with respect to ACE-FTS where the bias is reduced from 2% to almost 0%, and a degradation with respect to MLS, where the bias increases from 3% for the previous version to 5% for the new version. In the UTLS, the comparison with ACE-FTS still demonstrates a clear improvement, while MLS agrees better with the old version at 12-14 km altitudes, and better with the new version at 10-12 km and 14-16 km altitudes.

To access the drift, the old and the new data were compared to a ground-based network of ozonesonde and lidar stations, and to satellite ozone data from Aura MLS instrument. Obtained MIPAS drifts relative to the ground-based network are not significant for either version. The ground-based network analysis alone does not allow a robust conclusion whether the old or new version is the more stable. In the satellite analysis, the pattern of significant negative drifts at heights over 30 km and going down to 20 km at mid-latitudes, which was observed in the old version, has completely disappeared in the new version. The appearance of some pixels with significant positive drift in the new version is tentatively attributed to the over-correction in non-linearity coefficients in Level 1B spectra, which is expected to be corrected in the upcoming version 8 of Level 1B MIPAS spectra. The results of network and satellite drift analysis are consistent: both indicate that the drifts of the new version are less negative / more positive nearly everywhere above 15 km.

The 10-years MIPAS ozone trends calculated from the old and the new data versions were compared. The new trends are closer to old drift-corrected trends than the old uncorrected trends were. From this, we conclude that the non-linearity correction performed on Level 1B data is successfull, and probably also applicable to other species where no easy drift-correction via comparison to external instruments is possible.

*Acknowledgements.*  The authors are thankful to the referees for the comments that helped to improve the paper. We also thank Anne Kleinert for helpful comments. This work was performed in the frame of European Space Agency (ESA) project Ozone_cci. All four MIPAS teams acknowledge ESA for providing MIPAS L1b data. The ACE mission is supported primarily by the Canadian Space Agency. Work at the Jet Propulsion Laboratory was performed under contract with the National Aeronautics and Space Administration.

The service charges for this open access publication have been covered by a Research Centre of the Helmholtz Association. We acknowledge support by Deutsche Forschungsgemeinschaft and Open Access Publishing Fund of Karlsruhe Institute of Technology.

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

Wagner, G. and Birk, M.: Improvement of DLR detector non- linearity characterization for MIPAS/Envisat. Support to MIPAS Phase E activities Technical Note, issue 1A, 29 June 2005.