# Peer review of "On the improved stability of the version 7 MIPAS ozone record"

_Atmospheric Measurement Techniques, 2017_

## Referee Comment (RC1) · Anonymous Referee #1 · 6 Dec 2017

This is a very brief paper that assesses the stability of the new version 7 MIPAS ozone data product, which incorporates new time dependent detector calibrations. This is an important improvement in the MIPAS record and it should be documented. This paper could eventually be suitable for publication in AMT; however the current state of the paper is seriously lacking in important details, as listed in the comments below, some of which I consider major and a barrier to understanding what has been done and interpreting the results. A second major point that needs to be addressed, is that the work is rightly motivated to assess/use MIPAS for long term trends; however, the effect of the correction on derived trends is not addressed. The authors should revise or update previously reported trends using MIPAS data or perform at least a basic trend analysis on the old and new data records.

[Figure]

Page 2-3: The description and brief discussion on the new ozone retrieval algorithm (section 2) needs more detail/rigor. These changes and potential improvements are an important part of understanding the changes to the ozone product and several questions arise. For example, the text implies the ECMWF 2D temperature fields are used in the forward calculation for the temperature retrieval. This doesn't make sense. Are they used as the a priori for the temperature retrieval, or they used instead of the retrieved temperature in the ozone retrieval? What is the impact of using WACCM CO2? How/why is it appropriate to fit the background up to 60 km? Why was it fit to only 33 km previously? What is the quantitative effect of the joint retrieval with H2O? How is it that the max vertical resolution (6 km) is less than the max sampling (8 km)? Is the FR processor unchanged? Have the averaging kernels shown in Fig 1 changed from the previous version in a substantial way? If so, this should be pointed out, otherwise maybe consider not including the figure.

How exactly is the collocated MIPAS profile shifted to "prolong" the sonde profile? (I think "extend" is a better word to use rather than prolong). Is there some averaging of the two in an overlap region? At which altitude does this become important in the comparisons in Fig 3? The smoothed sonde profile should be explained, and not just presented with "edge effects". Presumably it trails off near 30 km because the authors have assumed a vmr of zero above the sonde measurements when applying the averaging kernel?

Is this level of "improvement" in bias between versions really significant in the Boulder comparisons? What does the statement mean exactly that "all biases are significant at 2 sigma level"? If so, then what can be said about the increased bias between 15-18 km? Is this also significant?

The discussion around the change in bias with respect to ACE and MLS around the 35 km vmr peak should be edited for clarity. It seems in both cases, the bias has increased in value, but for ACE it started out negative and is now closer to zero and for MLS it started out positive and is now more positive. This is not really what is implied with the

current discussion and as it stands it is difficult to reconcile.

Figure 5 and surrounding discussion is hard to interpret without a quantitative explanation of the non-linear coefficients and how they are applied, i.e. is an offset of -2 nW/sr/cm2/cm-1 a substantial change or not? In all cases shown, the year to year variability in the NLC oscillates about a trend. Is there an explanation for this? Are the NLC changes linearly interpolated when applied to the data as implied by the plots, or fit to a line, or something else? Are these NLC changes at 16 km and 51 km typical or extreme? What about altitudes between 16 and 51 km, such as 20-40 km where the trend results are quite important? Are the changes always monotonic with time? Why is spectral band D always unchanging?

The multilinear regression model used to assess the drift should be explained briefly so that a reference to the previous papers isn't necessary for a basic understanding of what is done here. A plot of the fit and the residuals would be informative. How linear is the correction and/or the remaining drift? Is it a good assumption to model it as linear in the regression?

Technical corrections:

Page 2 line 6: phrase in the past tense, something like, "the aging of the instrument was observed to change the detector response"

Page 2 line 12: what is a "parent" dataset?

Page 2, line 29: What is the "nominal mode"?

Page 5, line 2: "contrast" is not the right word to use to describe the difference in averaging kernels

Page 8, line 7-9: not a well formed sentence

Page 9: line 2: revise the wording of "a lot fewer"

---

## Referee Comment (RC2) · Anonymous Referee #2 · 18 Dec 2017

"On the improved stability of the version 7 MIPAS ozone record" by Laeng et al.

Opening Remarks

In this paper the authors have attempted to assess the stability of a new Level 2 MIPAS ozone data product. Two significant changes have been made to the Version 5 MIPAS processing chain to construct the new Version 7 MIPAS ozone data product. Detector nonlinearity corrections have been implemented at the Level 0 to Level 1 stage and modifications have been made to infer Level 2 ozone from the MIPAS Level 1 spectral radiance measurements. The paper concludes that comparisons with satellite data suggest that MIPAS Version 7 is more stable than MIPAS Version 5 and can be used within analyses that attempt to derive long-term atmospheric change. Although I believe the authors to be correct the paper does not provide near enough detail to make

a definitive statement. The paper appears to have been rushed and is not complete.

Comments

Major detail is lacking from the paper. A full validation study does not need to be done within the scope of this paper, but this document needs to address, in a more systematic way, the impact of the changes made to the MIPAS processing chain.

1) Section 2 that addresses algorithmic changes needs to be expanded. Sensitivity studies need to be presented that detail the impact of each of the outlined changes. If the current author list does not consist of a member of the retrieval team, then the list should be expanded so this work can be presented within this paper. Without more detail in this section it should be removed, and the paper is already far too brief.

2) I assume that Figure 1 was included to indicate that diagnostics such as averaging kernels are available. The figure needs far more explanation to be useful.

3) Section 2 was far too rushed. I am not suggesting it needs to be the length of a full paper. However, it needs to be expanded by two or three pages, with figures, to allow the reader to assess the changes made in the new Level 1 to Level 2 retrieval.

4) Section 3 on bias estimation also needs more detail. I feel that Figure 3 is telling me that the V7R bias has been reduced with respect to that of V5R. However, I don't believe the statistics associated with the comparison have been reported. For example, how many profiles went into the sonde study? Is the comparison data set presented in Figure 4 global and if so how relevant are the results? How many profiles went into calculating the results in Figure 4 and are there latitudinal biases that cancel each other out?

5) I don't believe section 3 is a bias estimation. It can be more correctly categorized as a sanity check that no additional biases have been added by the new processing. There are two issues with this. If the section is meant to be a bias estimation it must be expanded with a more comprehensive set of comparisons. The second issue is a

section titled bias estimation must be tied more closely to the title that is concerned with "stability". If this is not possible the title should be changed, or the section should be dropped. However, as previously mentioned the paper is already too short.

6) Section 4.2.1 needs much more detail. The authors reference the work done by Hubert et al. (2016) and present some sort of network averaged summary of work that is patterned after that of Hubert et al.. The authors need to take the time to repeat and present the entirety of the Hubert work, or as an alternative, come up with some better way of doing and presenting the same work. Without more detail this section should be removed from the paper as it adds nothing.

7) Section 4.2.2 is a good start. For the reader to trust that MIPAS stability has improved this section needs to be expanded. It needs to include a sensitivity analysis that indicates the methods outlined within Eckert et al. are appropriate. It may be as simple as just showing some representative time series where the drift is clearly seen and is clearly linear.

8) Section 4.2.2 would be well served by a direct comparison between V7 and V5. The same plot shown in Figure 7, as well as supporting figures, should be made for a direct V7 to V5 comparison.

9) The last thing that is missing is a trend analysis. What have the changes made to the Level 1 and Level 2 data products done to trends derived from MIPAS data? Do these now fit better with the accepted values seen in works like Steinbrecht et al. (2017)?

Summary

This paper is far too short to be accepted with minor revisions. Should the authors choose to address all the comments listed above I would be happy to review the completed manuscript. I hope they choose to make the suggested changes as the MIPAS data set provides very important information to the international teams involved in the determination and attribution of long term atmospheric change.

---

## Author Comment (AC1) · 27 Feb 2018

The authors thank the reviewers for the constructive comments which have helped to improve the paper.

**Answer to Reviewer #1.**

*This is a very brief paper that assesses the stability of the new version 7 MIPAS ozone data product, which incorporates new time dependent detector calibrations. This is an important improvement in the MIPAS record and it should be documented. This paper could eventually be suitable for publication in AMT; however the current state of the paper is seriously lacking in important details, as listed in the comments below, some of which I consider major and a barrier to understanding what has been done and interpreting the results.*

Please find our replies below

*A second major point that needs to be addressed, is that the work is rightly motivated to assess/use MIPAS for long term trends; however, the effect of the correction on derived trends is not addressed. The authors should revise or update previously reported trends using MIPAS data or perform at least a basic trend analysis on the old and new data records.*

Done. MIPAS 10-years trends were calculated in (Eckert et al, 2014) papier. One should be aware that trends in their paper are already drift-corrected. So we do not expect an improvement of final drift-corrected trend estimates from (Eckert et al, 2014). Instead, we expect that the new trends (left panel of the figure below) are closer to drift corrected trends from Eckert's paper (middle panel in the Figure below, this is Fig. 15 from Eckert's paper) than uncorrected original trends were (right panel in the figure below, this is Fig. 13 in Eckert's et al, 2014). As this expectation is confirmed, we learn that the non-linearity correction is ok and probably also applicable to other species where no easy drift-correction via comparison to external instruments is possible. The plot on left panel is now included into the paper as well as corresponding discussion.

[Figure]

V7R, no drift correction        V5R, drift corrected        V5R , no drift correction

*Page 2-3: The description and brief discussion on the new ozone retrieval algorithm (section 2) needs more detail/rigor. These changes and potential improvements are an important part of understanding the changes to the ozone product and several questions arise.*

The following discussion is now included into the paper:

"All improvements reported below refer both to the FR and the RR data. One of the most relevant changes in the improved retrieval setup for the IMK/IAA Level 2 Scientific processor is not related to the ozone retrieval itself but to the preceding temperature retrieval, of which the results are used in the ozone retrieval. The radiative transfer forward calculation now uses information on the along-line-of-sight horizontal two-dimensional temperature variation from the European Centre for Medium-Range Weather Forecasts (ECMWF) re-analysis. While the temperature at the measurement geolocation is still retrieved from the MIPAS spectra, the horizontal temperature variation in terms of temperature differences is constrained by the analysis data. This allows a more accurate retrieval in situations where neither the assumption of local horizontal uniformity nor the approximation of horizontal temperature variation by linear gradients, as described by (von Clarmann et al, 2009) is

adequate. Typical cases when the old approaches lead to problems are situations where the line of sight of a measurement crosses the edge of the polar vortex.

Above the uppermost tangent altitude, MIPAS cannot measure altitude-resolved temperature profiles any more. Since above the uppermost tangent altitude at most one degree of freedom is available, the a priori temperature profile is shifted there. In previous data versions, MSIS (Hedin, 1991) temperatures served as a priori at these altitudes. Particularly in situations of elevated stratopause events, this climatological model did not represent the actual upper stratospheric/lower mesospheric conditions very well, and the related temperature retrieval error was found to propagate noticeably onto the ozone concentrations retrieved. Validation results of the old version were found to be particularly poor during such elevated stratopause events. Thus, in the new version, a priori information on temperature at altitudes above 60 km is derived from the Whole Atmosphere Chemistry Climate Model (WACCM) temperature fields (Marsh et al., 2013). Also the $CO_2$ mixing ratios used for the temperature retrieval are now based on WACCM.

Further changes in the retrieval setup refer to the background continuum radiation. While this was previously considered only up to 33 km altitude, it is now fitted up to altitudes of 60 km. (Haenel et al, 2015) and (Plieninger et al, 2015) have shown that constraining the background continuum to zero above 33 km is inadequate and leads to artefacts in the retrieval. These improvements in the temperature retrieval indirectly affect the ozone retrieval, because the retrieved temperatures are used in the latter.

Although $H_2O$ is retrieved prior to ozone and any $H_2O$ signal in the ozone microwindows should thus be correctly accounted for, inconsistencies in spectroscopic data between $H_2O$ transitions in the microwindows used for the $H_2O$ retrieval and those in the $O_3$ microwindows might trigger an ozone retrieval error. Thus, although already known, $H_2O$ mixing ratios are treated as a fit variable in the ozone retrieval again. For this purpose two additional $H_2O$ microwindows have been included in the ozone retrieval in order to better constrain $H_2O$ in the upper troposphere and lower stratosphere (UTLS). The initial $H_2O$ result was used as a priori of the $H_2O$ joint fit during the $O_3$ retrieval.

Beyond this, the forward model KOPRA (Stiller, 2000) is now run with higher numerical accuracy. The number of pencil beams used for the numerical integration of radiances over the instrument fields of view is increased."

*For example, the text implies the ECMWF 2D temperature fields are used in the forward calculation for the temperature retrieval. This doesn't make sense. Are they used as the a priori for the temperature retrieval, or they used instead of the retrieved temperature in the ozone retrieval?*
This is now thoroughly described in the new text above.

*What is the impact of using WACCM CO2?*
This is now described above

*How/why is it appropriate to fit the background up to 60 km?*
References to Plieninger and Haenel are now given. Text has been rewritten, see above.

*Why was it fit to only 33 km previously?*
Because aerosol densities, which are thought to be the major cause of the continuum radiation up there, were thought to be too low up there to have any noticeable effect. This view, however, was erroneous.

*What is the quantitative effect of the joint retrieval with H2O?*
The description is now added into the text.

***How is it that the max vertical resolution (6 km) is less than the max sampling (8 km)?***
This wording was indeed sloppy, we changed it.

***Is the FR processor unchanged?***
The equivalent changes has been applied to FR processor, this is now added into the text.

***Have the averaging kernels shown in Fig 1 changed from the previous version in a substantial way? If so, this should be pointed out, otherwise maybe consider not including the figure.***

[Figure]

**V5R**                              **V7R**

The averaging kernels did change in a substantial way (see Figure above). We followed the suggestion and pointed this out, as well as the changes in the vertical resolution that it impled.

***How exactly is the collocated MIPAS profile shifted to "prolong" the sonde profile? (I think "extend" is a better word to use rather than prolong). Is there some averaging of the two in an overlap region?***
No averaging is performed. Typically the ozonesonde profiles are going up to 30 km heights. For each ozonesonde profile, at this uppermost height, the difference with value of collocated MIPAS profile at the same height was calculated. Then MIPAS profile at the upper height was glued to the ozonesonde profile, which provide a "profile" going up to 70 km.
We changed the wording from "prolong" to "extend".

***At which altitude does this become important in the comparisons in Fig 3?***
At the uppermost height of the ozonesonde profile.

***The smoothed sonde profile should be explained, and not just presented with "edge effects".***
Below is the original version of this plot, which was simplified in order not to overcharge the plot. Please see the discussion below.

[Figure]

*Presumably it trails off near 30 km because the authors have assumed a vmr of zero above the sonde measurements when applying the averaging kernel?*

No such assumption was made. Instead of assuming VMR zero, the ozonesonde profiles VMR were completed by the values of MIPAS collocated profile (green curve at the plot above), which was shifted (by the difference of green curve value at black curve value at height 29 km in this case) . The shifting is done in order to be glued with ozonesonde at its uppermost height (at the plot above, the resulting curve is violet). This is a standard procedure to overcome the edge effects when smoothing the ozonesonde profiles.

*Is this level of "improvement" in bias between versions really significant in the Boulder comparisons? What does the statement mean exactly that "all biases are significant at 2 sigma level"? If so, then what can be said about the increased bias between 15-18 km? Is this also significant?*

As initially suggested by the Editor, we changed this plot by including the error bars, and corresponding discussion was modified as well, including the significance of the bias aspect. According to the suggestion of the Review 2, this analysis is now resented as a sanity check, aiming to demonstrate that no artificial biases were introduced by the new processing scheme.

*The discussion around the change in bias with respect to ACE and MLS around the 35 km vmr peak should be edited for clarity. It seems in both cases, the bias has increased in value, but for ACE it started out negative and is now closer to zero and for MLS it started out positive and is now more positive. This is not really what is implied with the current discussion and as it stands it is difficult to reconcile.*

The discussion is revised.

*Figure 5 and surrounding discussion is hard to interpret without a quantitative explanation of the non-linear coefficients and how they are applied, i.e. is an offset of -2 nW/sr/cm2/cm-1 a substantial change or not? In all cases shown, the year to year variability in the NLC oscillates about a trend. Is there an explanation for this? Are the NLC changes linearly interpolated when applied to the data as implied by the plots, or fit to a line, or something else? Are these NLC changes at 16 km and 51 km typical or extreme? What about altitudes between 16 and 51 km, such as 20-40 km where the trend results are quite important? Are the changes always monotonic with time? Why is spectral band D always unchanging?*

The section was completely rewritten.

*The multilinear regression model used to assess the drift should be explained briefly so that a reference to the previous papers isn't necessary for a basic understanding of what is done here. A plot of the fit and the residuals would be informative.*

This model was explained in Eckert et al 2014 paper. We added a brief summary and extended the discussion.

*How linear is the correction and/or the remaining drift? Is it a good assumption to model it as linear in the regression?*

We see from the Fig 5 that the linearity is a good assumption. A corresponding discussion was added.

*Technical corrections:*
*Page 2 line 6: phrase in the past tense, something like, "the aging of the instrument was observed to change the detector response"*
*Page 2 line 12: what is a "parent" dataset?*
*Page 2, line 29: What is the "nominal mode"?*
*Page 5, line 2: "contrast" is not the right word to use to describe the difference in averaging kernels*

*Page 8, line 7-9: not a well formed sentence*
*Page 9: line 2: revise the wording of "a lot fewer"*
All rephrased, necessary explanations added.

**Answer to Reviewer #2.**

*In this paper the authors have attempted to assess the stability of a new Level 2 MIPAS ozone data product. Two significant changes have been made to the Version 5 MIPAS processing chain to construct the new Version 7 MIPAS ozone data product. Detector nonlinearity corrections have been implemented at the Level 0 to Level 1 stage and modifications have been made to infer Level 2 ozone from the MIPAS Level 1 spectral radiance measurements. The paper concludes that comparisons with satellite data suggest that MIPAS Version 7 is more stable than MIPAS Version 5 and can be used within analyses that attempt to derive long-term atmospheric change. Although I believe the authors to be correct the paper does not provide near enough detail to make a definitive statement. The paper appears to have been rushed and is not complete.*

*Comments*

*Major detail is lacking from the paper. A full validation study does not need to be done within the scope of this paper, but this document needs to address, in a more systematic way, the impact of the changes made to the MIPAS processing chain.*

*1) Section 2 that addresses algorithmic changes needs to be expanded. Sensitivity studies need to be presented that detail the impact of each of the outlined changes. If the current author list does not consist of a member of the retrieval team, then the list should be expanded so this work can be presented within this paper. Without more detail in this section it should be removed, and the paper is already far too brief.*
The relevant authors are already included. The section was rewritten; please see the answer to the Reviewer 1.

*2) I assume that Figure 1 was included to indicate that diagnostics such as averaging kernels are available. The figure needs far more explanation to be useful.*
The text was changed according to the suggestion of the Reviewer 1: substantial change in the AK as well as the implied changes in the vertical resolution are now reported and discussed. Please see the answer to the Reviewer 1 for illustration.

*3) Section 2 was far too rushed. I am not suggesting it needs to be the length of a full paper. However, it needs to be expanded by two or three pages, with figures, to allow the reader to assess the changes made in the new Level 1 to Level 2 retrieval.*
Done, please see answer to the Reviewer 1.

*4) Section 3 on bias estimation also needs more detail. I feel that Figure 3 is telling me that the V7R bias has been reduced with respect to that of V5R. However, I don't believe the statistics associated with the comparison have been reported. For example, how many profiles went into the sonde study? Is the comparison data set presented in Figure 4 global and if so how relevant are the*

***results? How many profiles went into calculating the results in Figure 4 and are there latitudinal biases that cancel each other out?***

The text was completely rewritten, we agree that the discussion was misleading.

The major bias reduction has already been achieved in the previous data version, reported in (Laeng et al, 2014). From improvements which are relevant in special situations only, we do not expect a general bias reduction. The text was adjusted accordingly.

***5) I don't believe section 3 is a bias estimation. It can be more correctly categorized as a sanity check that no additional biases have been added by the new processing. There are two issues with this. If the section is meant to be a bias estimation it must be expanded with a more comprehensive set of comparisons. The second issue is a section titled bias estimation must be tied more closely to the title that is concerned with "stability". If this is not possible the title should be changed, or the section should be dropped. However, as previously mentioned the paper is already too short.***

We rewritten the chapter, referenced our recent bias validation work (Laeng et al, 2015), and present this analysis now, as suggested, as a sanity check which shows that no unreasonable bias has been introduced by the changes in the processing setup.

***6) Section 4.2.1 needs much more detail. The authors reference the work done by Hubert et al. (2016) and present some sort of network averaged summary of work that is patterned after that of Hubert et al..The authors need to take the time to repeat and present the entirety of the Hubert work, or as an alternative, come up with some better way of doing and presenting the same work. Without more detail this section should be removed from the paper as it adds nothing.***

The added a summary of the work of Hubert et al. We however disagree that the section adds nothing: the stability of an instrument should be checked against satellite as well as ground based data records. The fact that ground-based analysis alone does not allow us to conclude if the stability was improved does not mean that the analysis is meaningless.

***7) Section 4.2.2 is a good start. For the reader to trust that MIPAS stability has improved this section needs to be expanded. It needs to include a sensitivity analysis that indicates the methods outlined within Eckert et al. are appropriate. It may be as simple as just showing some representative time series where the drift is clearly seen and is clearly linear.***

The section was rewritten, a sensitivity analysis included.

***8) Section 4.2.2 would be well served by a direct comparison between V7 and V5. The same plot shown in Figure 7, as well as supporting figures, should be made for a direct V7 to V5 comparison.***

Such a plot is done and is now included into the paper.

***9) The last thing that is missing is a trend analysis. What have the changes made to the Level 1 and Level 2 data products done to trends derived from MIPAS data?***

We included now a plot as in Fig. 13 in Eckert paper. Please see the answer to the Reviewer 1 for the discussion of what could one expect from a 10-years only trend analysis.

***Do these now fit better with the accepted values seen in works like Steinbrecht et al. (2017)?***

MIPAS record is 10 years long, which is not long enough to fit the solar cycle. In order to make a meaningful trend estimation, it should be merged with previous and post-MIPAS data records. In Steinbrecht et al 2017, the MIPAS V7 record was merged with SAGE II and OMPS. Comparing the

values from 10-years fit with values from 35 years fit does not make sense to us. Instead, we compare the obtained 10-years trends from MIIPAS with previous non-corrected and drift-corrected trends from Eckert et al, 2014 (see plot below), we observe that the new trends are closer to the drift-corrected trends from Eckert that to their non-corrected trends, and conclude from this that the drift-correction in V7 was successful.

[Figure]

V7R, no drift correction          V5R, drift corrected          V5R , no drift correction

***Summary***
***This paper is far too short to be accepted with minor revisions. Should the authors choose to address all the comments listed above I would be happy to review the completed manuscript. I hope they choose to make the suggested changes as the MIPAS data set provides very important information to the international teams involved in the determination and attribution of long term atmospheric change.***

---

## Author Response (AR2)

**Answer to the Report 2**

The authors thank the reviewer for the helpful comments.

*The revision of this paper is substantially improved and has met the concerns of the original reviews. The only remaining substantial issue is the organization of the paper (plus a few minor points and spelling and grammar mistakes that the authors should attend to).*

*In terms of organization, the early sections of the paper are laid out in a confusing order. Figure 2 is called out before Figure 1, and the discussion surrounding Figure 1 comes much later. Then the effect of the changes in the retrieval on ozone trends are discussed immediately at the end of section 2.1 before anything to do with ozone trends, including methodology, are mentioned and/or referenced (this comes much later in section 4). I realize that they are addressing two different issues (retrieval updates and level 1 spectra updates), however the authors need to review the overall story of the paper once more and organize appropriately.*

We reorganized the paper accordingly.

*Two other smaller issues:*
*(1) The claim made at the end of section 2.3 on the "clear improvement" in the UTLS. This is certainly not clear from the figure. More philosophically, if a profile has both high and low biases at various altitudes and a change is made that results in a simple shift of the entire profile (as seems to be the case here), is it fair to say that the regions now closer to zero are improved?*

If the new bias profile was shifted, we would agree that one can obviously not speak about improvement. However, at Figure 7, green and blue curves are not shifted, there are points where they coincide (22 km, 29 km, 42 km), and there are regions, where the green curve (corresponding to new version) is closer to zero. We argue that this is an improvement.

*(2) The claim that the drift is "clearly linear" at the end of section 3.2 is unsubstantiated.*
We changed it to "approximately linear".

*Corrections:*
*- Page 3, line 25: "change as to 2D field"*
*- Page 4, line 12: reference CCMI-REFC1SD*
*- Page 5, line 5: "compared i climatological"*
*- Page 7, line 5: "desactivated"*
*- Page 7, line 11: "inidividual"*
*- Page 7, line 18: rephrase "non-trustful" here and elsewhere*
*- Page 7, line 19: "The is no"*
*- Page 10, line 8: "are not be visible"*
*- Page 13, line 6: "see the Figure 9"*
*- Page 14, line 21: not a sentence*
*- Page 17, line 16: "access" is not a good choice of wording here*

All done